# Immune Checkpoint Inhibitors for Metastatic Colorectal Cancer: A Systematic Review

**DOI:** 10.3390/cancers17132125

**Published:** 2025-06-24

**Authors:** Alice Gilson, Vincent Tan, Thibaud Koessler, Jeremy Meyer, Guillaume Meurette, Émilie Liot, Frédéric Ris, Vaihere Delaune

**Affiliations:** 1Division of Digestive Surgery, Geneva University Hospitals, 1205 Geneva, Switzerland; jeremy.meyer@hug.ch (J.M.); guillaume.meurette@hug.ch (G.M.); emilie.liot@hug.ch (É.L.); vaihere.delaune@hug.ch (V.D.); 2Internal Medicine, La Tour Hospital, 1217 Geneva, Switzerland; vincent.tan@latour.ch; 3Oncology Department, Geneva University Hospitals, 1205 Geneva, Switzerland; thibaud.kossler@hug.ch; 4Hepatology and Transplantation Laboratory, Department of Surgery, Faculty of Medicine, University of Geneva, 1205 Geneva, Switzerland

**Keywords:** colorectal cancer, immunotherapy, checkpoint inhibitor, microsatellite stability (MSS), microsatellite instability (MSI), cytotoxic T lymphocyte antigen 4 protein (CTLA-4), programmed cell death protein 1 (PD-1), programmed death-ligand 1 (PD-L1)

## Abstract

Colorectal cancer presents a widespread and significant health concern, prompting extensive exploration into innovative therapeutic strategies. Immunotherapy has emerged as a promising field of research in this pursuit. By employing specialized proteins, immunotherapy aims to enhance the body’s innate immune response against cancerous cells. This study provides an overview of the current evidence regarding immune checkpoint inhibitors in metastatic colorectal cancer, assessing whether recent studies could impact current treatment guidelines. Recent studies have demonstrated the potential effectiveness of monoclonal antibodies that target specific molecular markers associated with colorectal cancer. Ongoing investigations are also examining the combined effects of immunotherapy with conventional non-surgical treatments, such as chemotherapy, radiotherapy, antiretroviral agents, or metformin, aiming to optimize patient outcomes. However, further research is needed to gain full understanding of the therapeutic benefits of immunotherapy and its potential as a first-line treatment option for colorectal cancer.

## 1. Introduction

Colorectal cancer (CRC) is the third most prevalent cancer worldwide, with nearly 2 million new diagnoses in 2020 [1]. It is the second leading cause of cancer-related death, accounting for almost one million deaths annually [2,3]. Despite improvements in surgical and oncological management, survival for patients with metastatic disease remains poor. In stage IV colon cancer, a 5-year survival of 13% was reported between 2013 and 2019 [4].

Immunotherapy has emerged as a promising therapeutic strategy across several malignancies, including melanoma, non-small cell lung cancer, and more recently, CRC. Its rationale in colorectal cancer is based on the ability of certain tumors to evade immune surveillance by exploiting immune checkpoint pathways. In particular, tumors with mismatch repair deficiency (dMMR) or high microsatellite instability (MSI-H) harbor a high mutational burden, resulting in the production of neoantigens that make them more recognizable by the immune system. These tumors are more likely to respond to immune checkpoint blockade.

Among the various immunotherapeutic approaches, monoclonal antibodies—especially immune checkpoint inhibitors—have shown the most promise in CRC. These antibodies work by modulating the immune response through targeting specific protein interactions [5]. The most frequently used monoclonal antibodies for colorectal cancer are pembrolizumab and nivolumab, which target the programmed cell death protein 1 (PD-1) receptor, avelumab, and durvalumab, which target programmed cell death protein-ligand 1 (PD-L1), and ipilimumab, which targets cytotoxic T lymphocyte-associated protein 4 (CTLA-4).

Immune checkpoint inhibitors act by releasing the physiological “brakes” that prevent T-cell activation. Under normal conditions, checkpoint pathways such as PD-1/PD-L1 and CTLA-4 serve to limit autoimmunity and excessive immune activation. Tumor cells often exploit these pathways to suppress anti-tumor immune responses. PD-1 is an inhibitory receptor expressed on T-cells, and its engagement by PD-L1 or PD-L2 on tumor or antigen-presenting cells leads to T-cell exhaustion. Anti-PD-1 (e.g., pembrolizumab or nivolumab) and anti-PD-L1 (e.g., avelumab or durvalumab) antibodies block this interaction, restoring T-cell activity. CTLA-4, on the other hand, competes with the costimulatory receptor CD28 for binding to B7 molecules on antigen-presenting cells. By inhibiting CTLA-4, these drugs enhance early T-cell priming and expansion. The net effect of these agents is to reinvigorate T-cell-mediated cytotoxicity against tumor cells.

While other types of immunotherapy exist [6], including cytokine-based therapies [7], adoptive cell therapy [8], cancer vaccines [9], or virotherapy [10], this systematic review will focus on monoclonal antibodies due to the very limited data available on the other treatments.

This systematic review provides an up-to-date summary of recent studies on the efficacy and toxicity of checkpoint inhibitors in the treatment of metastatic colorectal cancer.

## 2. Materials and Methods

This systematic review was conducted in accordance with the Preferred Reporting Items for Systematic Reviews and Meta-Analyses (PRISMA) 2020 guidelines [11]. The review protocol was prospectively registered in the Research Registry (reviewregistry1897) on 7 October 2024.

A comprehensive literature search was performed independently by two reviewers (VT, AG) across three major databases: MEDLINE, Web of Science, and Cochrane. The search strategy used the following MeSH terms: (1) “colorectal cancer” AND “immunotherapy” and (2) “colorectal cancer” AND “checkpoint inhibitor”. Filters were applied to identify randomized controlled trials (RCTs) and clinical trials published between January 2019 and January 2025.

The primary outcomes of interest were progression-free survival (PFS), overall survival (OS), and objective response rate (ORR). To ensure inclusivity of relevant studies, the term “colorectal cancer” was used without distinction between colon and rectal cancer, reflecting the nomenclature in the majority of eligible publications.

All titles and abstracts were screened, and duplicates were removed prior to full-text review. Studies were excluded if they involved vaccination immunotherapy, cytokine-based therapy, adoptive cell therapy, cancer vaccines, or virotherapy due to insufficient clinical data. Additional exclusion criteria included studies on non-metastatic colorectal cancer, pre-clinical models, in vitro experiments, investigations focusing exclusively on predictive biomarkers or histological analyses, as well as protocols and non-RCT/non-clinical trial designs.

Although a meta-analysis was initially considered, substantial heterogeneity among studies with respect to intervention types, treatment regimens, patient populations, and outcome reporting precluded meaningful statistical synthesis. As a result, no meta-analysis was conducted.

## 3. Results

### 3.1. Inclusion Process

We identified three hundred and forty-nine studies through a literature search, from which fifty-five duplicates were removed. Based on the title/abstract and full-text review, we excluded a further two hundred forty-one studies: Of these, eighty-nine studies were unrelated to immunotherapy in colorectal cancer, thirty focused on immunotherapy vaccines or another type of immunotherapy, twenty-seven focused solely on predictive factors of treatment response or histology analysis, ninety-two were at protocol stage or not RCT/clinical trials, and three were on population with locally advanced but not metastatic colorectal cancer. We therefore included fifty-three studies. Figure 1 displays the PRISMA flow chart detailing the process of inclusion and exclusion in our search. While most of the studies were about colorectal cancer only, some also conducted cross-sectional research on different cancers.

### 3.2. Immune Checkpoint Inhibitor Monotherapy

Checkpoint inhibitors targeting PD-1 and PD-L1 have differing efficacy in metastatic colorectal cancer (mCRC), depending on the microsatellite status of the tumor. See Table 1.

#### 3.2.1. Anti-PD1 Monotherapy

The KEYNOTE-177 trial demonstrated a clear advantage of pembrolizumab as a first-line treatment in microsatellite instability-high (MSI-H) mCRC. It significantly improved the objective response rate (ORR) and doubled progression-free survival (PFS) compared to standard chemotherapy. Overall survival (OS) was also increased. However, despite this encouraging result, superiority was not demonstrated due to a predefined α of 0.025 for statistical significance. Nevertheless, a high crossover rate (60%) from the chemotherapy arm to subsequent anti-PD-1 therapy likely diluted the OS benefit, making it difficult to reach statistical significance despite a real clinical advantage. This trial also highlighted pembrolizumab’s favorable safety profile, with fewer high-grade treatment-related adverse events (TRAEs) [12].

Iparomlimab, a newly developed humanized anti-PD-1 checkpoint inhibitor, was tested as a monotherapy and demonstrated promising anti-tumor effects, with a clinically significant and durable response in the mCRC subset of patients, as well as an acceptable safety profile. A confirmatory clinical trial, aiming to enroll at least 200 patients with MSI solid tumors, is currently underway [14].

Cetrelimab, a fully human immunoglobulin G4 monoclonal anti-PD-1 antibody, was investigated in pretreated patients (at least one previous treatment regimen), with a subgroup analysis indicating better response in the MSI-H group. However, the outcome was poor with more than 50% grade 3 or above TRAEs [15].

#### 3.2.2. Anti-PD-L1 Monotherapy

Anti-PD-L1 monotherapies have also demonstrated their efficacy in MSI-H CRC. When used in mCRC MSI patients progressing after first-line therapy, avelumab improved PFS but did not improve ORR compared to standard second-line chemotherapy. It did, however, induce less grade 3 or above TRAEs [16].

Envafolimab is a humanized single-domain PD-L1 antibody fused to an Fc fragment, allowing for a subcutaneous administration. Its therapeutic efficacy and safety profile were tested in advanced solid MSI-H tumors. The ORR was 28.0% for the CRC subgroup, the median PFS was estimated at 7.2 months, and the median OS was not reached. TRAEs of grade 3–4 were reported in 16% of cases, and no grade 5 adverse events were observed, with a low incidence of injection site reactions [17].

The effectiveness of immune checkpoint inhibitors in microsatellite stability (MSS) mCRC remains minimal. The IMblaze370 trial, assessing atezolizumab with or without cobimetinib, demonstrated poor outcomes in terms of ORR and PFS compared to the multikinase inhibitor, regorafenib [18]. Survival rates were generally low across all treatment arms, contrasting with findings in preclinical mouse models [19,20,21]. Furthermore, atezolizumab alone was met with a high percentage of TRAEs.

### 3.3. Immune Checkpoint Inhibitor (ICI) Combinations

Combining two immune checkpoint inhibitors (ICIs) offers a theoretical advantage by targeting complementary pathways to enhance anti-tumor immune responses, potentially overcoming resistance mechanisms and improving clinical outcomes. See Table 2.

#### 3.3.1. Anti-PD-1 and Anti-CTLA-4 Combinations

The Checkmate 142 trial assessed a combination of nivolumab with low-dose ipilimumab in MSI-H mCRC. This approach demonstrated robust clinical benefits. Despite a median follow-up exceeding two years, progression-free and overall survival had yet to be reached, reflecting the treatment’s sustained effectiveness. The safety profile remained manageable, with a limited proportion of patients experiencing high-grade treatment-related adverse events [22].

Similarly, the Checkmate 8HW trial, a phase 3 trial, confirmed the significant efficacy of this drug combination in MSI-H mCRC, showing a 24-month PFS of 72% versus 14% with standard of care chemotherapy, with a favorable safety profile [23].

For MSS mCRC, the combination of botensilimab and balstilimab showed encouraging clinical activity, suggesting potential benefits even in a setting typically resistant to immunotherapy [24]. However, toxicity was more pronounced, with a notable proportion of patients experiencing severe adverse events, particularly in the 3-week interval group [25]. Despite the absence of a control group, these findings highlight the promise of novel immunotherapeutic strategies in MSS mCRC, while underscoring the need for careful management of treatment-related toxicities.

#### 3.3.2. Anti-PD-L1 and Anti-CTLA-4 Combinations

Tremelimumab combined with durvalumab was tested in MSS CRC patients with liver metastases, with results showing limited differences compared to supportive care. Threefold higher rates of grade ≥ 3 TRAEs were observed in the intervention group [26].

In a different study, neoadjuvant tremelimumab plus durvalumab, followed by adjuvant durvalumab alone, was tested in mCRC patients with resectable liver metastases prior to CRC liver metastasis resection. Included patients either had MSS, MSI, or POLE mutated tumors. The median PFS was 9.7 months, and OS reached 24.5 months. Four patients (17%) showed a complete pathological response, 50% of these patients were MSI, and the other 50% were ultramutated cancers with a POLE mutation [27].

#### 3.3.3. A Novel Dual-Target ICI and a Bispecific Antibody

To address the significant toxicity associated with the combination of anti-PD-1 and anti-CTLA-4 antibodies, which have been extensively studied in clinical trials, a novel agent, QL1706, was developed. This bispecific antibody integrates two monoclonal antibodies in one compound: one targeting PD-1 and the other CTLA-4. Notably, the CTLA-4 component has been engineered for a shorter half-life, aiming to reduce immune-related toxicities. While this approach has shown encouraging response rates and durable disease control in metastatic or recurrent cancers, its effectiveness in mCRC appears limited, and tumor microsatellite status was not specified in the trial [28].

Similarly, ivonescimab, a bispecific antibody targeting both PD-1 and vascular endothelial growth factor (VEGF), has demonstrated a favorable safety profile and promising activity across multiple solid tumors. However, its impact on mCRC remains unclear, as there was only a small subset of patients within the study cohort, limiting the strength of conclusions [29].

### 3.4. Checkpoint Inhibitors Combined with Other Monoclonal Antibodies

Combinations of checkpoint inhibitors with monoclonal antibodies have been explored in metastatic colorectal cancer (mCRC) to enhance immune activation and overcome resistance observed with monotherapies. See Table 3.

#### 3.4.1. Anti-PD1 with Other Monoclonal Antibodies

The glucocorticoid-induced tumor necrosis factor receptor (GITR) plays an important role in regulating the immune system. It is mainly expressed in T lymphocytes. Activation of the GITR receptor has several effects on immunity, such as reducing the suppression of Tregs, promoting the proliferation and survival of effector T-cells, and enhancing their effector function. Investigations into targeting GITR with the agonist antibody MK-1248 showed an 18% ORR when combined with pembrolizumab, compared to an ORR of 0% in monotherapy [30].

Ziv-aflibercept, an anti-angiogenic agent, has been explored in combination with pembrolizumab for six patients. While this approach demonstrated some disease control, the overall impact on survival was modest [31].

Bevacizumab is an anti-VEGF (vascular endothelial growth factor) monoclonal antibody. Combined with pembrolizumab and binimetinib (a small molecule inhibitor), the study tried to assess the ability of MAPK and VEGF pathway blockade to overcome resistance to immunotherapy in microsatellite stable metastatic colorectal cancer. However, it failed to meet its primary endpoint of higher ORR compared with historical control data [32].

Sabatolimab, a monoclonal antibody targeting T-cell immunoglobulin domain and mucin domain-3 (TIM-3), showed a partial response of 6% when combined with spartalizumab (anti-PD1) in unspecified microsatellite status mCRC patients, with a recommended phase 2 trial dose of 800 mg [33].

NIS793 is a human IgG2 monoclonal antibody blocking transforming growth factor beta (TGF-β), a key regulator of the tumor microenvironment involved in fibroblast activation, immune exclusion, and suppression. Combining NIS793 with spartalizumab allowed for a stabilization of the disease in 24.2% of MSS mCRC and a partial response in 3.5% of patients. However, further studies are warranted to confirm this data [34].

TAS-116, an oral HSP90 inhibitor that inhibits the activity of regulatory T-cells in peripheral blood mononuclear cells and tumor-infiltrating lymphocytes, combined with nivolumab was disappointing, with 100% tumor progression under this regimen at data cutoff, despite having relatively few TRAEs [35].

A combination of pixatimod, a TLR9 activator, and nivolumab was assessed in MSS mCRC. The TLR9 (toll-like receptor) activates PRR pathways, stimulating tumor-resident innate immune cells like dendritic cells. This activation releases cytokines, upregulates costimulatory molecules, and promotes cross-priming of tumor antigens. This combination showed 12% partial response and 32% stable disease [36].

#### 3.4.2. Anti-PDL1 with Other Monoclonal Antibodies

As mentioned previously, atezolizumab with or without cobimetinib demonstrated poor results in terms of ORR and PFS in MSS CRC, compared with the multikinase inhibitor regorafenib [18], Table 1.

The Morpheus-CRC trial, combining atezolizumab with isatuximab (an anti-CD38 antibody), also showed poor results, with a lack of overall response and shorter overall survival (OS) in the intervention arm (5.1 vs. 10.2 months) [37].

Because CD73 upregulation in tumors leads to local immunosuppression, oleclumab, an anti-CD73 human IgG1λ monoclonal antibody, was assessed in combination with durvalumab versus oleclumab alone in a cohort of patients with MSS cancers, of which 21.9% had mCRC. The combination had little effect in the mCRC subgroup, with only a 5.4% median PFS at 6 months and an OS of 7 months. Treatment had very few TRAEs [38].

Monalizumab is an antiNKG2A/CD94 that may promote anti-tumor immunity by targeting innate immunity. A combination with durvalumab, to target both innate and adaptive immunity, in patients with advanced solid tumors showed a good tolerance and evidence for immune activation; however, clinical impact was minimal, with an ORR of 7.7% in the mCRC subgroup [39].

### 3.5. Immune Checkpoint Inhibitors Combined with Conventional Treatments

#### 3.5.1. Checkpoint Inhibitors with Chemotherapy

Microsatellite stable (MSS) colorectal cancers have a notoriously mediocre response to immune checkpoint inhibitors (ICIs) alone. ICIs have therefore also been studied in combination with conventional chemotherapy, exploring possible synergistic effects. See Table 4.

##### Anti-PD1 with Chemotherapy

Pembrolizumab was tested in combination with modified FOLFOX6, the current established treatment for mCRC, in patients with no prior systemic therapy. The observed PFS with this combination was 8.8 months [40]. Despite validating the safety of this regimen, with 30% of grade 4 TRAEs, pembrolizumab had no added value as the results were similar to those obtained with modified FOLFOX6 alone [70].

Another study attempted to mitigate the lack of response of MSS mCRC to pembrolizumab monotherapy by combining it with azacitidin in patients who had already been treated with standard chemotherapy. However, this study was prematurely terminated due to the absence of clinical response [41].

The Checkmate 9 × 8 trial found that nivolumab added to standard-of-care chemotherapy (SOC, 5-fluorouracil/leucovorin/oxaliplatin/bevacizumab) in predominantly MSS mCRC patients and increased ORR in the intervention group (60% vs. 46%). However, PFS was similar in both groups at 11.9 months. The combination increased the incidence of grade ≥ 3 TRAEs, observed in 75% of patients versus 48% in the SOC group [42].

The METIMOX trial was also underwhelming. The combination of nivolumab and FLOX had a worse outcome than FLOX alone, with an ORR of 47% vs. 65%. Furthermore, grade ≥ 3 TRAEs were more frequent in the intervention arm (26% vs. 14%) [43].

The CAROSELL trial tested nivolumab combined with CDX101 (a histone deacetylase inhibitor) in MSS mCRC. The combination achieved a partial response rate of 9% and stable disease in 39% of patients, resulting in an ORR of 9%. Despite acceptable TRAEs, outcomes were worse than those reported in other studies with the standard of care [44].

Lastly, nivolumab combined with TAS-102 (trifluridine/tipiracil) was assessed in MSS mCRC patients. This trial was prematurely terminated due to lack of efficacy, and significant toxicity [45].

Serplulimab was combined with the chemotherapy regimen of HLX04 (bevacizumab biosimilar) and XELOX in predominantly MSS mCRC patients. This combination increased PFS to 17.2 months compared to 10.7 months in the control group (placebo + bevacizumab + XELOX group), with similar OS and with similar rates of TRAEs [46].

##### Anti-PDL1 with Chemotherapy

Atezolizumab combined with FOLFOXIRI and bevacizumab in previously untreated mCRC improved PFS compared to the control group, with similar safety profiles [47,48].

In MSS or MSI-L mCRC, the addition of avelumab to the standard regimen of mFOLFOX6 and cetuximab showed promising response rates, with a high proportion of patients achieving objective responses. The progression-free survival observed in this combination was notable and came with a favorable safety profile [49]. However, there was no significant additional benefit compared to historical data on mFOLFOX4 + cetuximab [71,72].

#### 3.5.2. Checkpoint Inhibitors with Small Molecule Inhibitors

Small molecule inhibitors can modify cancer cell signaling pathways in ways that increase their susceptibility to immune system-mediated destruction. Combining them with checkpoint inhibitors has shown potential for synergistic effects with promising results in murine models, thereby spurring recent clinical investigations [73].

##### Anti-PD1 with Small Molecule Inhibitors

The combination of pembrolizumab and ibrutinib, a Bruton’s tyrosine kinase (BTK) inhibitor, was tested in refractory MSS mCRC patients who were intolerant or unresponsive to standard chemotherapy. The treatment showed little to no anti-cancer activity [50].

The LEAP-017 trial assessed pembrolizumab combined with lenvatinib, a multikinase inhibitor, compared with the standard of care (SOC) in MSS or MSI-low CRC. The combination resulted in an improved ORR compared to the SOC group. Median PFS and OS were slightly increased, with similar TRAEs [51].

Navarixin is a small molecule inhibitor designed to target the CXC chemokine receptor 2 (CXCR2). CXCR2 plays a role in inhibiting cell proliferation in normal cells; however, in the tumor microenvironment, its presence paradoxically leads to increased tumor cell proliferation. Navarixin was tested in combination with pembrolizumab in multiple solid tumors, including mCRC, but the trial was terminated at interim analysis for lack of efficacy [52].

The combination of nivolumab and ipilimumab with regorafenib has been assessed in two different trials, showing encouraging results (ORR of 27.6% and 36.4%, median PFS of 4 months and 5 months, and median OS of 20 and 27.5 months, respectively) [53,54]. The discrepancy between the two studies is likely due to the inclusion of only non-liver metastasic patients in the second study. A subgroup analysis of non-liver metastatic patients in the first study showed similar outcomes.

The combination of nivolumab alone with regorafenib was slightly less effective, comparable to literature-reported standard-of-care results [55,56]

In MSS mCRC patients with BRAF V600E mutation, a combination of encorafenib, a small molecule BRAF inhibitor, with nivolumab and cetuximab, reported encouraging results, with very little TRAEs [57]. These results have led to the initiation of an ongoing follow-up phase II randomized trial [74].

The combination of regorafenib and toripalimab reported an ORR of 15.2% with a median PFS of 2.1 months. Subgroup analysis found an ORR of 30% in non-liver metastatic patients versus 8.3% in liver metastatic patients [58].

##### Anti-PDL1 with Small Molecule Inhibitors

Durvalumab was tested in combination with trametinib, a MEK inhibitor, in MSS mCRC. This combination did not show a significant change in PFS, and the ORR was limited to 3.4%. Among the responders, one partial response lasted for 9.3 months [59].

It also has been tested with olaparib (PARP inhibitor) or cediranib (VEGFR inhibitor) in a randomized trial, but both combinations showed limited anti-tumor activity [60].

Finally, a combination of durvalumab with cabozantinib, a small molecule tyrosine-kinase inhibitor, demonstrated some anti-tumor activity, notably on PFS, and manageable toxicity, leading to the development of a phase III trial [61].

The MODUL trial explored various targeted therapies across distinct biomarker-driven subgroups of mCRC patients. Treatment included vemurafenib, cetuximab, and 5-FU/LV for BRAF-mutated tumors; capecitabine, trastuzumab, and pertuzumab for HER2-mutated tumors; and cobimetinib with atezolizumab for MSI-H or MSS tumors harboring BRAF or RAS mutations. While no significant differences in PFS were observed between treatment arms, the study highlighted a higher prevalence of mitogen-activated protein kinase (MAPK) pathway mutations in the intervention group, which may have negatively influenced treatment outcomes [62].

#### 3.5.3. Checkpoints Inhibitors with Radiotherapy

Given that neither immune checkpoint inhibition alone nor radiotherapy alone achieves a sufficient effect in MSS mCRC, a possible solution would be to combine both treatments. Radiotherapy can promote the release of tumor antigens through cancer cell death, thereby alerting the immune system and attracting immune cells to the tumor, and immunotherapy could reinforce this immune response by increasing lymphocyte activity and promoting the recognition and elimination of cancer cells [75].

A combination of durvalumab, tremelimumab, and radiotherapy was assessed in 24 chemotherapy refractory MSS mCRC patients. Unfortunately, this trail failed to meet the expected endpoints (defined as ≥ 3 of 24 objective responses by RECIST v1.1) and was not considered for further study. Interestingly, however, some rare cases of increased systemic immunity and abscopal responses, characterized by the regression of non-irradiated lesions, were observed [63].

Building on the theoretical abscopal response effect and promising preclinical trials combining ipilimumab and nivolumab with radiotherapy, a trial explored the synergistic potential of these three treatments in improving systemic responses in MSS mCRC. The combination of ipilimumab and nivolumab was met with a high number of TRAEs, leading 33% of patients to discontinue treatment prior to radiation therapy. Nevertheless, for those who managed to follow through with radiotherapy, responses were observed particularly in terms of the duration of disease control. There is an intent to carry out a confirmatory phase 3 study to further investigate these findings [64].

#### 3.5.4. Checkpoints Inhibitors with Other Treatments

A novel immune sensitizing strategy was investigated in the MAYA trial, a phase II study assessing the efficacy of temozolomide priming followed by a combination of low-dose ipilimumab and nivolumab in patients with MSS mCRC. This approach was based on the rationale that temozolomide-induced DNA damage could enhance immunogenicity and increase susceptibility to immune checkpoint blockade. Only 33 of the included 135 patients completed both treatment parts, and 76% were discontinued because of disease progression or death during the priming phase. Of the 24% remaining who completed the whole protocol, the 8-month progression-free survival rate was 36%, with a median PFS of 7.0 months and an overall response rate of 45%. Despite the limited number of patients reaching the immune checkpoint blockade phase, the study provided proof-of-concept that a sequential approach with temozolomide priming could enhance the efficacy of immunotherapy in a subset of MSS mCRC patients [65].

Research has addressed the lack of response of MSS mCRC to pembrolizumab monotherapy, finding that adding maraviroc, an anti-HIV agent inhibiting C-C motif chemokine receptor 5 (CCR5), leads to an anti-tumor activation of macrophages. This phase 1 clinical trial achieved a feasibility rate of 94.7%, which reflects the proportion of patients who tolerated the treatment without experiencing severe adverse events leading to premature discontinuation. However, anti-tumor clinical response was low despite translational analyses revealing an increase in anti-tumor chemokines during treatment [66].

Another study investigated the effectiveness of immunotherapy in “cold” solid tumors, defined as tumors that lack tumor-infiltrating lymphocytes in their microenvironment, an indicator of non-response to immunotherapy. However, combining DNA hypomethylating agent CC-486 with durvalumab did not demonstrate significant clinical efficacy [67].

Inspired by preclinical research indicating that metformin reduces exhaustion of tumor-infiltrating lymphocytes and potentiates PD-1 blockade, the combination of nivolumab with metformin in MSS mCRC patients did not demonstrate efficacy despite being well tolerated by patients [68].

Finally, a study aimed to identify a novel combination therapy to overcome anti-PD-1 resistance, employing enadenotucirev, a tumor-selective blood-stable adenoviral vector, in combination with nivolumab in patients with advanced/metastatic epithelial cancer. Despite low clinically relevant effects, this study demonstrated that this combination can promote immune cell infiltration and activation, with increases in intra-tumoral CD8+ T-cells and cytolytic activity [69].

## 4. Discussion

Metastatic colorectal cancer (mCRC) remains a therapeutic challenge. This systematic review underscores the growing interest in using immune checkpoint inhibitors as potential alternative therapy. While checkpoint inhibitors (ICIs) have demonstrated significant efficacy in microsatellite instability-high (MSI-H) patients due to their high tumor mutational burden, these patients only account for 5% of advanced colorectal tumors [76]. Several challenges remain for expanding their utility to the 95% of remaining cases, the microsatellite-stable (MSS) mCRC, and optimizing patient outcomes across all mCRC subtypes.

The efficacy of checkpoint inhibitors combination as a first-line treatment option in MSI-H mCRC is well-documented, such as pembrolizumab in KEYNOTE-177 demonstrating marked improvements in PFS and ORR [12] and nivolumab with ipilimumab in Checkmate 142 or Checkmate 8HW [22,23]. Envafolimab is also promising in previously treated MSI-H mCRC. These treatments are also well tolerated with limited severe toxicity.

Thus far, the most promising results for MSS mCRC without liver metastases seem to be the combination of nivolumab with regorafenib and ipilimumab. However, most other therapeutic strategies, such as combining ICIs with other ICIs, monoclonal antibodies, or conventional treatment, have very little effect on MSS mCRC, while accumulating toxicity of the combined molecules. This highlights significant biological barriers, including the cold tumor microenvironment and limited immune infiltration, which continue to hinder therapeutic efficacy. Approaches such as modifying the tumor microenvironment through anti-TGF-β or anti-CD73 antibodies, activating innate immunity with TLR9 agonists, or employing combination regimens involving checkpoint inhibitors and targeted therapies show potential. Furthermore, current ongoing strategies to convert MSS tumors to MSI tumors could potentially expand the population of patients who can benefit from immunotherapy; of note, the recent MAYA trial, although met with a high number of patients with disease progression or death during the priming phase, showed promising results when patients completed the entire treatment protocol. Combining ICIs with chemotherapy, small molecule inhibitors, or radiotherapy represents a promising strategy to enhance immune activation and overcome resistance mechanisms in MSS CRC. However, despite relative safety and results that are statistically significant, the observed effects of currently available molecules and tested combinations have very little meaningful clinical impact, only increasing PFS, ORR, and OS by a couple of weeks or months.

While monotherapy with checkpoint inhibitors, particularly in MSI-H mCRC, demonstrated a favorable safety profile with relatively low rates of grade ≥ 3 treatment-related adverse events (TRAEs), combination regimens were consistently associated with increased toxicity. The most frequently reported TRAEs across studies included diarrhea, fatigue, rash, elevated liver enzymes, and immune-related events such as colitis, pneumonitis, and endocrinopathies (e.g., hypothyroidism and hypophysitis). In combinations involving anti-CTLA-4 agents like tremelimumab, higher rates of immune-related adverse events were observed compared to PD-1/PD-L1 monotherapy [26]. Regimens combining checkpoint inhibitors with chemotherapy or targeted therapies, such as regorafenib or lenvatinib, further increased toxicity risks, including hypertension, neutropenia, and palmar–plantar erythrodysesthesia [51,56]. Importantly, discontinuation rates due to toxicity varied but were notably high (>10%) in certain combination arms, particularly those using three-drug regimens [25,64]. Identifying predictive markers for both response and toxicity therefore remains critical to optimize patient selection and therapeutic value.

As currently available molecules have little impact on MSS mCRC, researchers are currently developing “anti-cancer” vaccines designed to specifically sensitize the immune system to recognize and attack cancer cells. These vaccines can be derived from either whole cancer cells, modified fragments of cancer cells, or unique proteins and carbohydrates found only on cancer cells, absent in healthy tissues. However, as this field of research remains in its infancy, we chose not to explore it within this review. Another emerging therapeutic approach is virotherapy, which utilizes engineered viruses to selectively target and destroy cancer cells. This strategy aims to achieve dual effects: producing proteins that stimulate an immune response and directly eliminating cancer cells. While promising, virotherapy is still under active investigation and has yet to reach widespread clinical application.

Although this review has exhaustively reported on the current state of knowledge on immunotherapy and metastatic colorectal cancer, there are some incompressible limitations to the current systematic review. The available evidence is heterogeneous, notably in terms of tumor profiles tested (MSS vs. MSI), the reporting of these tumor profiles, and the combinations of molecules studied. Many studies included in this review are constrained by small sample sizes, diverse patient populations, and lack of comparative control groups. Furthermore, despite knowing that colon and rectal cancers are two very different entities with different tumor biology, we did not distinguish between rectal and colon cancers instead using the umbrella term “colorectal cancer” in our literature search, as most studies do not yet differentiate between these entities.

Additionally, most of the reviewed studies did not provide detailed information about prior chemotherapy received by patients participating in the studies, which could impact the efficacy of the studied molecules. Moreover, most non-randomized clinical trials study therapeutic combinations in a variety of cancer types and do not always report their results according to cancer subgroups due to the low number of included patients. It is therefore difficult to interpret the cancer-specific efficacy in these studies.

## 5. Conclusions

The findings of this systematic review highlight the growing interest in immunotherapy in colorectal cancer (CRC), particularly for patients with microsatellite instability-high (MSI-H) metastatic disease. Immune checkpoint inhibitors (ICIs) have demonstrated significant efficacy in this subgroup, representing a major advancement in treatment.

However, the limited responsiveness of microsatellite-stable (MSS) CRC underscores persistent biological challenges, such as the immunosuppressive tumor microenvironment. In this context, combinatorial strategies for MSS mCRC have emerged as particularly promising, especially the combination of regorafenib with nivolumab or ipilimumab in patients without liver metastases, which showed encouraging activity in early-phase studies.

Despite these advances, several obstacles remain. The toxicity associated with combination therapies, while often manageable, calls for robust predictive tools to optimize patient selection and minimize adverse outcomes.

This review underscores the need for deeper exploration of underlying mechanisms, better patient stratification, better differentiation between tumor phenotype, and rigorous clinical trials to expand access and maximize the benefits of immunotherapy across all CRC subtypes.

## Figures and Tables

**Figure 1 cancers-17-02125-f001:**
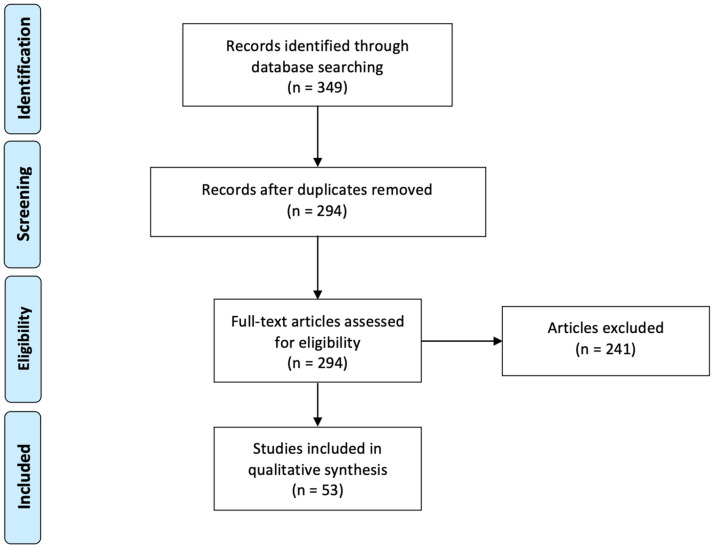
PRISMA flow diagram for studies included in this qualitative review.

**Table 1 cancers-17-02125-t001:** Immune checkpoint inhibitor monotherapy.

	Authors and Trial Name	Population	MSI/MSS Status (%)	Intervention	Control	Outcome (95% CI)	Safety/Adverse Events
Anti-PD-1	André et al. [12], Diaz et al. [13] **KEYNOTE-177**	*N* = 852 No prior treatment Randomized 1:1	100% MSI-H	Pembrolizumab (200 mg/3 weeks)	Chemotherapy repeated every 2 weeks	**ORR:** 43.8% (35.8–52) vs. 33.1% (25.8–41.1) **Complete response:** 11% vs. 4% **PFS:** 16.5 months (5.4–32.4) vs. 8.2 months (6.1–10.2) **OS**: not reached (49.2—NR) vs. 36.7 months (27.6—NR), ns	22% vs. 66% grade ≥ 3 TRAEs (including one patient who died in the chemotherapy group)
	Bi et al. [14]	*N* = 38 CRC patients out of a cohort of 60 patients	100% MSI	Iparomlimab 200 mg/3 weeks	-	**ORR:** 57.9% (52.9% for liver metastatic patients, 61.9% for non-liver metastatic patients) **PFS:** median PFS not evaluable, 63.4% at 24 months **OS:** median OS not evaluable, 65.5% at 24 months	20.8% grade ≥ 3 TRAEs
	Felip et al. [15]	*N* = 48 patients out of a cohort of 204 patients	56.3% MSI-H, 25% MSS, others non evaluable (insufficient tumor sample)	Cetrelimab 240 mg/2 weeks	-	**ORR**: 16.7%, 23.8% in MSI-H CRC **PFS**: 2.1 months **OS**: -	53.9% grade ≥ 3 TRAEs
Anti-PD-L1	Taieb et al. [16] **SAMCO-PRODIGE 54**	* N * = 122 Progression under standard first-line therapy Randomized 1:1	100% MSI	Avelumab (10 mg/kg/2 weeks)	Standard second-line chemotherapy ± targeted agent	**ORR**: 29.5% vs. 26.2% **Estimated PFS at 12 months:** 31.2% vs. 19.4% **OS**: 25.8 months (14.1-NR) vs. 23.4 months (13-NR)	31.7% vs. 53.1% ≥ 3 TRAEs
	Li et al. [17]	*N* = 65 CRC patients out of a cohort of 103 patients with previously treated solid tumors	100% MSI-H	Envafolimab (150 mg/week)	-	**ORR**: 43.1% (30.8–56.0) in the CRC subgroup **PFS**: 7.2 months for the CRC subgroup **OS**: 87.1% at 12 months for the CRC subgroup	16% grade ≥ 3 TRAEs
	Eng et al. [18] **IMblaze370**	* N * = 363	92% MSS 2% MSI	2 groups: Atezolizumab only Atezolizumab + cobimetinib	Regorafenib	**ORR**: 2%, 3%, and 2% **PFS:** 1.94, 1.91, and 2.00 months, respectively **OS:** 7.1, 8.9, and 8.1 months, respectively	61% and 31% vs. 58% ≥ 3 TRAEs

CRC: colorectal cancer—MSI: microsatellite instability—MSS: microsatellite stability—NR: not reached—NS: not significant—ORR: objective response rate—OS: overall survival—PFS: progression-free survival—TRAEs: treatment-related adverse events.

**Table 2 cancers-17-02125-t002:** Immune checkpoint inhibitor combinations.

	Authors and Trial Name	Population	MSI/MSS Status (%)	Intervention	Control	Outcome (95% CI)	Safety/Adverse Events
Anti-PD-1 and Anti-CTLA-4 Combinations	Lenz et al. [22] **Checkmate 142**	*N* = 45	100% MSI	Nivolumab every 2 weeks + low dose Ipilimumab every 6 weeks	-	**ORR**: 69% **PFS and OS**: not reached with minimum follow-up of 24.2 months	22% grade ≥ 3 TRAEs
	André et al. [23] **Checkmate 8HW**	*N* = 303 Randomized in 2:1 ratio	84% MSI	Nivolumab (240 mg) + Ipilimumab (1 mg/kg) each 3 weeks	Chemotherapy with or without targeted therapies	**PFS at 24 months:** 72% vs. 14%	23% grade ≥ 3 TRAEs vs. 48%
	El-Khoueiry et al. [24]	*N* = 59	100% MSS	Botensilimab 1 or 2 mg/kg every 6 weeks + balstilimab 3 mg/kg every 2 weeks	-	**ORR**: 22% **PFS**: - **OS**: 61% at 12 months, median OS not reached	34% grade ≥ 3 TRAEs
	Bullock et al. [25]	*N* = 148 All pre-treated	100% MSS	Botensilimab (every 3 or 6 weeks) + balstilimab 3 mg/kg every 2 weeks for 2 years	-	**ORR**: 17% (10–26%) **PFS**: 3.5 months (2.7–4.1) **OS**: 20.9 months (10.6-NR)	32% grade 3–4 TRAEs No grade 5 TRAE 12% discontinuation rate when botensilimab was given/3 weeks
Anti-PD-L1 and Anti-CTLA-4	Chen et al. [26]	*N* = 180 70.5% with liver metastasis All received all available standard systemic therapies	1.1% MSI 92.2% MSS	Tremelimumab + durvalumab during a median of 12 weeks	Best supportive care	**PFS:** 1.8 vs. 1.9 months **OS:** 6.6 vs. 4.1 months	64% vs. 20% ≥ grade 3 TRAEs
	Marie et al. [27]	*N* = 24	87.5% MSS 8.3% MSI 4.2% POLE	Tremelimumab and durvalumab 1 dose + adjuvant durvalumab	-	**ORR**: - **PFS**: 9.7 months **OS**: 24.5 months	22% grade ≥ 3 TRAEs
Dual-target ICI; bispecific Antibody	Zhao et al. [28]	*N* = 27 CRC patients out of a cohort of 518 patients Metastatic or recurrent	Unspecified	QL1706 (bifunctional PD1/CTLA4 dual blocker) at one of five doses ranging from 0.3 to 10 mg/kg, once every 3 weeks	-	**ORR**: 7.4% in CRC, 16.9% in the whole cohort **PFS**: - **OS**: - **Median duration of response**: 11.7 months for the whole cohort	16% grade ≥ 3 TRAEs
	Frentzas et al. [29]	*N* = 9 CRC patients out of a cohort of 51 patients	100% MSS	Ivonescimab (bispecific monoclonal antibody anti-PD1 and anti-VEGF) 0.3, 1, 3, 10, 20, or 30 mg/kg IV every 2 weeks using a 3 + 3 + 3 dose escalation design	-	**ORR**: 25.5% **PFS**: - **OS**: -	27.5% grade ≥ 3 TRAEs

CRC: colorectal cancer—MSI: microsatellite instability—MSS: microsatellite stability—ORR: objective response rate—OS: overall survival—PFS: progression-free survival —TRAEs: treatment-related adverse events.

**Table 3 cancers-17-02125-t003:** Checkpoint inhibitors combined with other monoclonal antibodies.

	Authors and Trial Name	Population	MSI/MSS (%)	Intervention(s)	Control	Outcome	Safety/Adverse Events
Anti-PD1 with other monoclonal antibodies	Geva et al. [30]	*N* = 8 CRC patients out of a cohort of 37 patients Group A&B: *N* = 20 (4 CRC) Group D: *N* = 17 (4 CRC)	Unspecified	Group D: MK-1248 + pembrolizumab	Group A&B: MK-1248 (anti-GITR antibody) alone	**ORR**: 18% vs. 0% Partial response: 12% vs. 0% Complete response: 6% vs. 0% No PFS or OS	45% on monotherapy group vs. 53% on combination therapy grade ≥ 3 TRAEs
	Rahma et al. [31]	*N* = 6	100% MSS	Ziv-aflibercept + pembrolizumab		**PFS**: 2.5 months **OS**: 3.3 months	58% grade ≥ 3 TRAEs
	Lentz et al. [32]	*N* = 50	100% MSS	Pembrolizumab + binimetinib + bevacizumab	-	**ORR:** 12.0% (not statistically different than the historical control data of 5%) **PFS**: 5.9 months **OS**: 9.3 months	64% grade ≥ 3 TRAEs
	Curigliano et al. [33]	*N* = 219 133 with Sabatolimab alone 86 with Sabatolimab + spartalizumab All had progressed under or were intolerant to standard therapy	Unspecified	Sabatolimab + spartalizumab	Sabatolimab alone	Partial response: 0% Sabatolimab alone 6% lasting 12–27 months with combination of Sabatolimab + Spartalizumab No ORR, PFS, or OS	51% grade ≥ 3 TRAEs
	Bauer et al. [34]	*N* = 53 CRC patients out of a cohort of 120 patients 60 patients in dose escalation group, 60 in dose expansion group	100% MSS for the 53 CRC	NIS793 +/- spartalizumab		3.5% partial response, 24.2% stable disease, 55.8% disease progression, and 17.5% unreported responses **PFS**: 1.41 months, regardless of whether patients received single-agent or combination No ORR or OS	99.2% TRAE of any grade, 57.5% TRAE grade ≥ 3
	Kawazoe et al. [35] EPOC1704	*N* = 29 CRC patients out of a cohort of 44 patients 57% received three or more previous lines of chemotherapy	97% MSS, 3% MSI	TAS-116 + nivolumab	-	**ORR**: 16% in CRC subgroup **PFS**: 3.2 months in CRC subgroup **OS**: 13.5 months in CRC subgroup	27% grade ≥ 3 TRAEs
	Lemech et al. [36]	*N* = 25 CRC patients out of a cohort of 58 patients	100% MSS	3 + 3 dose escalation Pixatimod + nivolumab	-	12% partial response, 32% stable disease No ORR, PFS, or OS	21% TRAE grade ≥ 3, 12% in CRC subgroup
Anti-PDL1 with other monoclonal antibodies	Chen Li et al. [37] **MORPHEUS-CRC**	*N* = 28 + 28 from the Imblaze370 study data	100% MSS	Atezolizumab + isatuximab	Regorafenib	**ORR:** 0% vs. 0% **PFS:** 1.4 months vs. 2.8 months **OS:** 5.1 months vs. 10.2 months	13% vs. 70% of grade ≥ 3 TRAEs
	Bendell et al. [38]	*N* = 42 CRC patients out of a cohort of 192 patients 66 during escalation and 126 during expansion	100% MSS	Oleclumab + durvalumab	Oleclumab monotherapy	**ORR**: - **PFS**: 1.8 months in both, median PFS of 5.4% at 6 months in CRC subgroup **OS**: 5.6 months in the combination therapy cohort, 6.1 months in the monotherapy cohort, and 7 months for CRC subgroup	19% TRAE grade ≥3 in CRC subgroup
	Patel et al. [39]	*N* = 40 CRC patients out of a cohort of 140 patients	100% MSS-CRC	Durvalumab 1500 mg + monalizumab 750 mg	-	**ORR**: 7.7% in CRC subgroup **PFS**: 1.9 months in CRC subgroup **OS**: 10.6 months in CRC subgroup	5% grade 3/4 TRAEs in CRC subgroup

CRC: colorectal cancer—MSI: microsatellite instability—MSS: microsatellite stability—ORR: objective response rate—OS: overall survival—PFS: progression-free survival —TRAEs: treatment-related adverse events.

**Table 4 cancers-17-02125-t004:** Immune checkpoint inhibitors combined with conventional treatments.

	Authors	Population	MSI/MSS (%)	Intervention	Control	Outcome	Safety/Adverse events
Checkpoint inhibitors with chemotherapy	Herting et al. [40] **MK-3475**	*N* = 6	66% MSI, 33% unassessed	Pembrolizumab + modified FOLFOX6	-	**ORR**: 56.7% **PFS**: 8.8 months **OS**: not reached as median follow-up was 19.9 months	30% grade 4 TRAEs
	Kuang et al. [41]	*N* = 30	Not reported	Pembrolizumab + azacitidine	-	**ORR**: 3% **PFS**: 1.9 months **OS**: 6.3 months	2.5% grade ≥ 3 TRAEs
	Lenz et al. [42] **Checkmate 9X8**	* N * = 195 Randomized 2:1	93% MSS	Nivolumab + SOC (5-fluorouracil/leucovorin/oxaliplatin/bevacizumab) (*N* = 127)	SOC (*N* = 68)	**ORR**: 60% vs. 46% **PFS**: 11.9 months vs. 11.9 months **OS:** 29.2 months vs. not reached	75% vs. 48% ≥ 3 TRAEs
	Ree et al. [43] **METIMMOX**	* N * = 80 Randomized 1:1	100% MSS	Nivolumab + FLOX	FLOX alone	**ORR:** 47% vs. 65% **PFS:** 9.2 months vs. 9.2 months If CRP < 5.0 mg/L before Nivolumab (*N* = 17) **PFS** 15.8 months **OS:** 20.7 months vs. 14.6 months	26% vs. 14% grade ≥ 3 TRAEs
	Saunders et al. [44] **CAROSELL**	*N* = 55	100% MSS	Nivolumab + CDX101 (zabadinostat)	-	**ORR** 9% achieved partial response and 39% stable disease **PFS**: 2.1 months **OS**: 7.0 months	25% grade ≥3 TRAEs
	Patel et al. [45]	*N* = 18	100% MSS	Nivolumab + TAS-102 (trifluridine/tipiracil)	-	**ORR**: 0% **PFS**: - **OS**: - stopped at stage I because no partial or complete response	72% grade ≥ 3 TRAEs
	Wang et al. [46]	* N * = 114 No prior therapy Randomized 1:1	95.7% MSS	Serplulimab + HLX04 + XELOX	Placebo + bevacizumab + XELOX	**PFS**: 17.2 vs. 10.7 months **OS**: not reached in either group	65.5% vs. 56.1% ≥ 3 TRAEs
	Antoniotti et al. [47,48] **AtezoTRIBE**	*N* = 218 Randomized 2:1 All previously untreated	100% MSS	Atezolizumab + FOLFOXIRI + bevacizumab (*N* = 145)	FOLFOXIRI + bevacizumab (*N* = 73)	**PFS:** 13.1 months vs. 11.5 months **OS: -** **ORR: -**	27% vs. 26% ≥ grade 3 TRAEs
	Stein et al. [49]	*N* = 43	100% MSS or MSI-low	Avelumab + mFOLFOX6 + cetuximab	**-**	**ORR**: 81% **PFS**: 11.1 months	32% grade ≥ 3 infections and neutropenia
Checkpoints inhibitors with small molecule inhibitors	Kim et al. [50]	*N* = 40	100% MSS	Pembrolizumab + ibrutinib	**-**	**ORR**: 0% **PFS**: 1.4 months **OS**: 6.6 months	40% grade ≥ 3 TRAEs
	Kawazoe et al. [51] **LEAP-017**	* N * = 480 Randomized 1:1	100% MSS or MSI-L	Pembrolizumab + lenvatinib	SOC	**ORR**: 10.4% vs. 1.7% **PFS**: 3.8 vs. 3.3 months **OS**: 9.8 vs. 9.3 months	58.4% vs. 42.1% ≥ 3 TRAEs
	Armstrong et al. [52]	*N* = 40 CRC patients out of a cohort of 105 patients	100% MSS	Pembrolizumab (200 mg/3 weeks) + navarixin (30 mg or 100 mg daily)	**-**	**ORR**: 2.5% for CRC subgroup **PFS**: 1.8 months for CRC subgroup with 30 mg and 1.9 months for CRC subgroup with 100 mg **OS**: 6.5 months (subgroup 30 mg) and 8.0 (subgroup 100 mg)	25% in navarixin 30 mg and 22% in navarixin 100 mg grade ≥ 3 TRAEs
	Fakih et al. [53]	*N* = 39	100% MSS	Nivolumab + regorafenib + ipilimumab	**-**	**ORR**: 27.6% **PFS**: 4 months **OS**: 20 months	37.9% grade ≥ 3 TRAEs
	Xiao et al. [54]	*N* = 22	100% MSS	Nivolumab + regorafenib + ipilimumab	**-**	**ORR**: 36.4% **PFS**: 5.0 months **OS**: 27.5 months	25% of patients discontinued treatment because of TRAEs but there is no clear number of grade ≥3 TRAEs
	D.Kim et al. [55]	*N* = 52	100% MSS	Nivolumab + regorafenib	**-**	**ORR**: - **PFS**: 4.3 months **OS**: 11.1 month	51% grade ≥ 3 TRAEs
	Kukuoka et al. [56] **REGONIVO, EPOC160**	*N* = 25 CRC patients out of a cohort of 50 patients All had received >2 previous lines of chemotherapy	96% MSS, 4% MSI	Nivolumab + regorafenib	**-**	**ORR**: 36% in CRC **PFS**: 7.9 months in CRC **mOS**: not reached in CRC **One-year OS**: 68.0% in CRC	40% grade ≥ 3 TRAEs
	Morris et al. [57]	*N* = 26	100% MSS	Encorafenib (300 mg PO daily) + cetuximab C (500 mg/m^2^ IV/14 days) + nivolumab (480 mg IV/28 days)	-	**ORR**: 45% **PFS**: 7.3 months **OS**: 11.4 months	18% grade ≥ 3 TRAEs
	Wang et al. [58]	*N* = 42	100% MSS 50% with RAS mutation 4.8% with BRAF mutation	Regorafenib + toripalimab	-	**ORR**: 15.2% Patients with liver metastasis had a lower response rate (ORR 8,7% vs. 30%) **PFS**: 2.1 months **OS**: 15.5 months	38.5% grade 3 TRAEs, no grade 4 or 5
	Johnson et al. [59]	*N* = 29	100% MSS	Durvalumab + trametinib	**-**	**ORR**: 3.4% **PFS**: 3.2 months **OS**: The one partial response lasted for 9.3 months	8% grade ≥ 3 TRAEs
	Hernando-Calvo [60]	*N* = 31	100% MSS	Durvalumab + olaparib	Durvalumab + cediranib	**ORR:** 0 **PFS:** 2.6 and 2.4 months, respectively, in CRC patients **OS**: 6.6 months for the two groups of CRC patients	28% grade 3 TRAEs, no grade 4
	Saeed et al. [61] **CAMILLA CRC cohort**	* N * = 29	100% MSS	Durvalumab + cabozantinib	-	**ORR**: 27.6% **PFS**: 44.83% **OS**: 9.1 months	39% grade ≥ 3 TRAEs
	Ducreux et al. [62] **MODUL trial**	Group 1: *N* = 60 Group 3: *N* = 5 Group 4: *N* = 99	Group 1: BRAF mutated Group 3: HER2+ Group 4: MSI-H or MSS witBRAFmut or RASmut	. Group 1: Vemurafenib + cetuximab + 5-FU/LV . Group 3: Capecitabine + trastuzumab + pertuzumab . Group 4: Cobimetinib + atezolizumab	Fluoropyrimidine + bevacizumab	**PFS:** no difference **OS:** numerically, but not significantly, longer in the intervention arm	60.0% vs. 27.8% ≥ 3 TRAEs
With radiotherapy	Segal et al. [63]	*N* = 24 Chemotherapy-refractory MSS mCRC	100% MSS	Durvalumab + tremelimumab + Radiotherapy	**-**	**ORR**: 8.3% **PFS**: 1.8 months **OS**: 11.4 months	25% grade ≥ 3 TRAEs 13% discontinued tremelimumab
	Parikh et al. [64]	*N* = 40 CRC patients out of a cohort of 65 patients	100% MSS	Ipilimumab + nivolumab + Radiotherapy		**ORR**: 10% (15% for the 27 patients with radiotherapy) in CRC subgroup **PFS**: 2.4 months in CRC subgroup **OS**: 7.1 months in CRC subgroup	70% grade ≥ 3 TRAEs in CRC subgroup 32% discontinued given autoimmune toxicity
With other treatments	Morano et al. [65] **MAYA Trial**	*N* = 135 started the first treatment part; 33 achieved the second treatment part	100% MSS	First part treatment: two cycles of oral temozolomide Second part: in absence of progression, ipilimumab and nivolumab	**-**	**ORR:** 45% **PFS:** 36% at 8 months **median PFS:** 7.0 months **OS**: 18.4 months	12% grade ≥ 3 TRAEs in the 33 patients (24.4%) who attained the second treatment part
	Haag et al. [66] **PICCASSO**	*N* = 20	-	Pembrolizumab + maraviroc	**-**	**ORR**: 5.3% **PFS**: 2.1 months **OS**: 9.83 months	5% grade ≥ 3 TRAEs
	Taylor et al. [67]	N = 28 (19 in A, 9 in B)	100% MSS	Group A: CC-486 + durvalumab Group B: CC-486+ durvalumab + vitamin C	**-**	**ORR**: 0% **PFS**: 1.9 months **OS**: 5 months	18% grade ≥ 3
	Akce et al. [68]	*N* = 18 Prior chemotherapy (5FU, Oxaliplatin, or Irinotecan)	100% MSS	Nivolumab + metformin	**-**	**ORR**: - **PFS**: 2.3 months **OS**: 5.2 months	44% grade ≥ 3 TRAEs
	Fakih et al. [69] **SPICE**	*N* = 45 CRC patients out of a cohort of 51 patients	All MSI-L or MSS	Nivolumab + enadenotucirev	**-**	**ORR**: 2.4% in CRC **PFS**: 1.6 months in total cohort (CRC subgroup not mentioned) **OS**: 16 months in total cohort (CRC not mentioned)	61% grade ≥ 3 TRAEs

CRC: colorectal cancer—MSI: microsatellite instability—MSS: microsatellite stability—ORR: objective response rate—OS: overall survival—PFS: progression-free survival—SOC: standard of care—TRAEs: treatment-related adverse events.

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
