# Peer review of "Immune Checkpoint Inhibitors for Metastatic Colorectal Cancer: A Systematic Review"

_cancers, 2025, doi:10.3390/cancers17132125_

Round 1

Reviewer 1 Report

Comments and Suggestions for Authors

The manuscript reviews metastatic colorectal cancer which is very challenging for treatment since 23% of colorectal cancer are diagnosed with metastatic cancer where 5-year relative survival is 16,2% (cancer.gov). The authors focused on the modern immunotherapy – immune check-point inhibitors in combination with other chemotherapeutics. They reviewed the outcomes of monotherapy with immune checkpoint inhibitors and 10 combinations including different immune checkpoint inhibitors and combinations with other types of chemotherapy and radiotherapy. The review will be very useful for chemotherapists treating colorectal cancers since each paragraph contains concise information about survival, adverse effects and molecular mechanisms of activity. The authors are also critical about the completeness of data they found in reviewed articles.

Author Response

Comment: 

The manuscript reviews metastatic colorectal cancer which is very challenging for treatment since 23% of colorectal cancer are diagnosed with metastatic cancer where 5-year relative survival is 16,2% (cancer.gov). The authors focused on the modern immunotherapy – immune check-point inhibitors in combination with other chemotherapeutics. They reviewed the outcomes of monotherapy with immune checkpoint inhibitors and 10 combinations including different immune checkpoint inhibitors and combinations with other types of chemotherapy and radiotherapy. The review will be very useful for chemotherapists treating colorectal cancers since each paragraph contains concise information about survival, adverse effects and molecular mechanisms of activity. The authors are also critical about the completeness of data they found in reviewed articles.

Response:

We thank the reviewer for their positive assessment and appreciation of the review’s structure and content. No specific revision was requested in this case.

Reviewer 2 Report

Comments and Suggestions for Authors

In my opinion, the submitted manuscript „ Immune Check-Point Inhibitors for Metastatic Colorectal Cancer: A Systematic Review” meets aims and scope of „Cancer” Journal, section Cancer Therapy, and Special Issue Diagnosis and Therapeutic Management of Gastrointestinal Cancers: 2nd Edition and may be accepted after the revision.

  1. I believe that the introduction of the publication could be expanded. In my opinion, more information is needed to understand why immunotherapy is applicable in the treatment of colon cancer and why it brings positive results.
  2. I believe that in the publication it would be appropriate to provide a deeper explanation of the mechanisms of action of individual antibodies or immune check-point inhibitors. Without deeper reflection, the publication may be perceived as merely describing what combinations of substances were used and whether they yielded positive effects or not.
  3. The publication requires the removal of excess dashes. They appear in many words and make reading difficult (see line: 28, 31, 57, 58, 59, 80, 86, 102, 104, 119, 120, 131, 135, 143, 156, 161, 166, 201, 206, 273, 296, 299, 301, 321, 374, 377, 385, 401, 422, 424, 429, 434, 452, 462, 469, 485, 486).
  4. The list of abbreviations could be presented in a more readable manner (perhaps as a list where one abbreviation is below the other, instead of one after the other in continuous text).

Author Response

Comment 1: The introduction could be expanded to better explain why immunotherapy is applicable in colon cancer and why it brings positive results.
Response 1: We agree and have expanded the introduction to better outline the rationale for using immunotherapy in CRC, especially in MSI-H tumors. We now provide more background on immune evasion, tumor immunogenicity, and why these mechanisms make checkpoint blockade relevant in this context. This has been updated in the introduction section.

Comment 2: Provide deeper explanation of the mechanisms of action of antibodies or immune checkpoint inhibitors.
Response 2: We have added a more detailed explanation of the mechanisms of action of PD-1, PD-L1, and CTLA-4 inhibitors, including their role in T-cell exhaustion and immune activation. This is now included in the Introduction section to strengthen mechanistic understanding for readers. 

Comment 3: Remove excess dashes that appear throughout the text 
Response 3: Thank you for this remark. We have thoroughly reviewed the manuscript and removed inappropriate dashes throughout the text to improve readability.

Comment 4: The list of abbreviations could be more readable (one per line rather than continuous text).
Response 4: We have reformatted the list of abbreviations into a vertical list, with one abbreviation per line, to enhance clarity and accessibility for the reader. 

Reviewer 3 Report

Comments and Suggestions for Authors

This systematic review comprehensively synthesizes recent evidence (2019-2025) on immune checkpoint inhibitors (ICIs) for metastatic colorectal cancer (mCRC), adhering to PRISMA guidelines. The inclusion of 48 studies focusing on PFS, OS, and ORR provides a robust update.

Suggestions for Improvement:

  1. Table Presentation: Table formatting (e.g., Table 1, Table 3) requires attention for consistency and readability (e.g., alignment, column headers spanning pages).

  2. Toxicity Depth: While TRAE rates are reported, a brief synthesis of common/notable toxicities across regimens in the results/discussion would be valuable.

  3. Conclusion Focus: The conclusion could more explicitly state the most promising combinatorial approaches for MSS mCRC (e.g., regorafenib/nivolumab/ipilimumab in non-liver mets) based on the presented data.

  4. Minor Typo: "anti-retroviral anti-HIV agent" (Simple Summary) is redundant; use "antiretroviral agents" or specify (e.g., maraviroc).

Author Response

Comment 1: Table formatting requires attention for consistency and readability 
Response 1: We have reviewed and corrected formatting inconsistencies in all tables. Column headers have been repeated across pages when tables span multiple pages. 

Comment 2: Toxicity Depth: While TRAE rates are reported, a brief synthesis of common/notable toxicities across regimens in the results/discussion would be valuable.
Response 2: We have added a dedicated paragraph in the Discussion section summarizing the most common and notable toxicities observed across the various regimens, including immune-related adverse events (e.g., colitis, hepatitis, endocrinopathies) and those related to specific combinations (e.g., palmar-plantar erythrodysesthesia with regorafenib). 

Comment 3: The conclusion could more explicitly state the most promising combinatorial approaches for MSS mCRC (e.g., regorafenib/nivolumab/ipilimumab in non-liver mets) based on the presented data.
Response 3: We have revised the conclusion to more clearly emphasize that, based on current evidence, the most promising approach in MSS mCRC is the combination of nivolumab, regorafenib, and ipilimumab in patients without liver metastases. 

Comment 4: Minor typo: "anti-retroviral anti-HIV agent" is redundant.
Response 4: Corrected as suggested. We now simply refer to “antiretroviral agent” in the Simple Summary.